# The Influence of Microstructure on TCR for Inkjet-Printed Resistive Temperature Detectors Fabricated Using AgNO_3_/Ethylene-Glycol-Based Inks

**DOI:** 10.3390/mi15060749

**Published:** 2024-06-02

**Authors:** Aziz Radwan, Yongkun Sui, Christian Zorman

**Affiliations:** 1Department of Electrical, Computer and Systems Engineering, Case Western Reserve University, Cleveland, OH 44106, USA; anr43@case.edu; 2Sandia National Laboratories, Albuquerque, NM 87123, USA; ysui@sandia.gov

**Keywords:** ethylene glycol, flexible sensors, temperature coefficient of resistance (TCR), inkjet printing, plasma reduction, printed sensors, silver ink, silver nitrate, resistance temperature detector

## Abstract

This study investigated the influence of microstructure on the performance of Ag inkjet-printed, resistive temperature detectors (RTDs) fabricated using particle-free inks based on a silver nitrate (AgNO_3_) precursor and ethylene glycol as the ink solvent. Specifically, the temperature coefficient of resistance (TCR) and sensitivity for sensors printed using inks that use monoethylene glycol (mono-EG), diethylene glycol (di-EG), and triethylene glycol (tri-EG) and subjected to a low-pressure argon (Ar) plasma after printing were investigated. Scanning electron microscopy (SEM) confirmed previous findings that microstructure is strongly influenced by the ink solvent, with mono-EG inks producing dense structures, while di- and tri-EG inks produce porous structures, with tri-EG inks yielding the most porous structures. RTD testing revealed that sensors printed using mono-EG ink exhibited the highest TCR (1.7 × 10^−3^/°C), followed by di-EG ink (8.2 × 10^−4^/°C) and tri-EG ink (7.2 × 10^−4^/°C). These findings indicate that porosity exhibits a strong negative influence on TCR. Sensitivity was not strongly influenced by microstructure but rather by the resistance of RTD. The highest sensitivity (0.84 Ω/°C) was observed for an RTD printed using mono-EG ink but not under plasma exposure conditions that yield the highest TCR.

## 1. Introduction

Resistive temperature detectors (RTDs) comprise the most commonly used class of temperature sensors due to their simple design, wide variety of scalable manufacturing methods, and high degree of reliability. Although RTDs can be fabricated using a wide range of approaches, additive manufacturing methods based on direct-write printing have emerged as a leading approach to fabricate these devices on flexible substrates for much the same reasons that make direct-write printing methods attractive for flexible electronics in general; namely, the high-quality metals used as interconnects and other conductive components in flexible electronics also exhibit the temperature-sensitive resistivity required for RTDs. From a commercial perspective, inkjet printing leads all methods of direct-write printing due to the availability of large-scale, high-throughput printing systems combined with a variety of commercially available inks. The most commonly used metal inks are based on silver (Ag) nanoparticle suspensions. To achieve a temperature coefficient of resistance (TCR) that is comparable to bulk Ag (3.8 × 10^−3^/°C), these processes require a thermal annealing step between 150 °C and 290 °C to sinter the printed structure [1,2,3,4,5,6,7,8]. Such high annealing temperatures preclude the use of temperature-sensitive substrates with these processes.

To expand the use of inkjet printing toward temperature-sensitive substrates, our group has been developing a process that bypasses the high-temperature annealing step required of Ag nanoparticle inks to form viable conducting structures. This process utilizes a low-pressure Ar plasma to form conductive Ag structures from particle-free inks comprised of silver nitrate (AgNO_3_) as the precursor and ethylene glycol as the ink solvent. During Ar plasma treatment, the substrate temperature is between ~39 °C and ~140 °C, depending on power and treatment time [9,10]. Such low substrate temperatures enable the fabrication of Ag devices on temperature-sensitive substrates, such as paper and cellophane. We have successfully used this approach to fabricate a wide range of Ag-based devices and sensors, including RC filters [11], RTDs [12], strain gauges [10], and chemical sensors [13]. For strain gauges, we found that the gauge factor exhibits a strong dependence on the microstructure of printed structures, which, in turn, depends on whether monoethylene glycol (mono-EG) or one of its derivatives, namely diethylene glycol (di-EG) or triethylene glycol (tri-EG), is used as the ink solvent. Likewise, we found that for chemical sensors, sensitivity was also strongly dependent on microstructure and, thus, the ink solvent. The aforementioned RTDs, however, were fabricated using only mono-EG ink. And while these RTDs exhibited an average TCR (1.91 × 10^−3^/°C) that compares favorably with its Ag nanoparticle counterparts as well as bulk Ag, the effect of microstructure on TCR using the broader family of ethylene-glycol-based inks is currently not known.

In this study, we expanded upon our previous work in developing particle-free AgNO_3_/ethylene glycol inks for temperature sensor applications by exploring the relationship between microstructure and TCR by characterizing the performance of RTDs fabricated using inks based on mono-EG, di-EG, and tri-EG solvents. We found that TCR is strongly affected by the porosity of the RTD, which, in turn, is strongly influenced by the type of ethylene glycol used as the ink solvent. To the best of our knowledge, a study focusing on the effect of microstructure on RTD performance has yet to be performed for any temperature sensors produced by inkjet printing. The AgNO_3_/ethylene glycol ink system is particularly well suited for such studies because a wide range of microstructures can be realized using ink solvents that are essentially of the same chemical behavior with respect to the Ag-containing precursor but behave differently during plasma treatment.

## 2. Experimental Details

### 2.1. Temperature Sensor Fabrication

Figure 1 schematically illustrates the ink preparation, printing, and plasma treatment steps used to fabricate the printed RTDs. AgNO_3_, a common inorganic silver-containing salt, was used as the metal precursor. The ink solvent consisted of deionized (DI) water mixed with either mono-EG, di-EG, or tri-EG. Three inks that differ only in the ink solvent were made: Ag/mono-EG, Ag/di-EG, and Ag/tri-EG, whose properties are listed in Table 1. The inks were prepared by mixing 1.7 g of AgNO_3_ (98% purity, Alfa Aesar Co., Ward Hill, MA, USA) with 16 mL of the glycol-based solvent and 4 mL of DI water in a vial in order to form inks with a molarity of 0.5 M. The mixture was then stirred for 15 min. The Ag salt was left to dissolve in the solvent for 24 h. DI water was used to match the viscosity and surface tension of the ink to the requirements of the inkjet printer.

RTDs were printed on cellophane tape substrates using a Dimatix DMP3000 series (FUJIFILM Dimatix, Inc., Lebanon, NH, USA) inkjet printer configured with a DMC-11610 print cartridge. The printer drop spacing was set at 15 μm, and the platen temperature was set at 50 °C for Ag/mono-EG, 43 °C for Ag/di-EG, and 41 °C for Ag/tri-EG inks to ensure the production of uniform patterns on the cellophane substrates. The heated platen facilitates evaporation of the DI water while leaving the EG solvent virtually unaffected. The nozzle voltage was set between 28 and 30 V. Multiple nozzles were activated simultaneously to print a single temperature sensor within 5 to 6 min. Cellophane tape (SCOTCH Matte-Finish ¾″ × 600″, 3 M, Maplewood, MN, USA) was selected as the substrate material due to its chemical compatibility with ethylene glycol (EG is used as a plasticizer in cellophane), favorable wetting properties of the inks, compatibility with the Ar-based plasma exposure process, and extensive prior experience by our group in using cellophane as a substrate material with these inks [11,12]. Moreover, the adhesive properties of cellophane tape make it an attractive substrate for inkjet-printed “stick-on” sensors. To facilitate handling, cellophane tape substrates were adhered to EPSON photo paper before printing. No special surface treatment steps were performed on the substrates prior to printing.

Immediately after printing, the as-printed samples were exposed to a low-pressure Ar plasma using a 300 W, 13.56 MHz, March PX250 plasma system (MKS Instruments, Milpitas, CA, USA). The plasma exposure step was performed in an Ar atmosphere at a pressure of 650 mTorr. The RF power was set at 150 W for samples printed using the mono-EG ink and 300 W for samples printed using the di-EG and tri-EG inks. Plasma exposure times ranged from 4 to 30 min in order to explore the effect of plasma exposure on sensor performance. A power of 300 W was selected for samples printed using the di-EG and tri-EG inks because previous work showed that a power of 150 W was insufficient to create structures with measurable electrical resistance even after 20 min of plasma exposure [10,13]. A power of 150 W was used for samples printed using mono-EG inks because previous work showed that 200 W produced structures with maximum conductivity in less than 15 min [11], which, for this study, was deemed too short to tune the resistance of printed structures using plasma exposure time as a control parameter. Previous work showed that for these plasma exposure conditions, the substrate temperatures range from 35 °C for an exposure time of 5 min at 150 W to 135 °C for an exposure time of 30 min at 300 W [18].

Figure 2 is a schematic diagram showing the plasma reduction steps associated with AgNO_3_/ethylene glycol [13]. During ink synthesis, AgNO_3_ dissolved in the glycol solvent dissociates into Ag^+^ and NO_3_^–^ ions and remains so throughout the printing step. During plasma exposure, electrons in the plasma dissolve in the ink and become solvated, which in turn react with Ag^+^ ions to form neutral Ag atoms. Neutral Ag atoms react within the ink to form solid Ag nuclei. Eventually, Ag nuclei coalesce to form solid structures. In other words, the Ag atoms sinter and form a conductive network.

Figure 3 presents a schematic diagram and representative photograph of a printed RTD after plasma exposure. These sensors consist of a single printed Ag layer configured in a conventional serpentine geometry with contact pads on each end of the structure. The width, length, and thickness of the sensor are 1 mm, 51 mm, and 1.5 µm, respectively. To achieve the target thickness, a fully formed RTD was printed after a single print pass.

### 2.2. Testing Setup

RTDs were evaluated using the custom-built setup detailed in Figure 4. The testing rig was composed of a transparent acrylic enclosure with circular through-holes for electrical connections to the temperature sensor and a strip heater. The strip heater (IC Station 12 W, 12 V Flexible Polyimide) was used for direct heating of the temperature sensors from room temperature to ~60 °C by placing the heater in direct contact with the temperature sensing element. To maximize heat conduction and minimize heat loss, the bottom of the temperature sensor was affixed to a thermally insulated glass slide, and the testing enclosure was filled with cotton. The resistance of the temperature sensor was monitored using a Fluke 79 III multimeter (Fluke, Everett, WA, USA), and temperature was monitored using a Teledyne FLIR C2 infrared thermal camera (Teledyne FLIR, Wilsonville, OR, USA).

### 2.3. Material Analysis

Plan-view scanning electron microscopy (SEM) using a Thermo Fisher Aprero 2 (Thermo Fisher Scientific, Waltham, MA, USA) was performed on a representative sensor printed using each ink to investigate the relationship between ink composition and the surface microstructure of each specimen. Previous work with these inks showed that plasma treatment led to the formation of polycrystalline Ag structures consisting of (111), (220), (220), and (311) grains [11].

## 3. Results

### 3.1. Resistance versus Plasma Exposure Time

Figure 5a presents the room temperature resistance versus plasma exposure time for RTDs printed using the mono-EG, di-EG, and tri-EG inks. For all three inks, the resistance decreased with increasing plasma exposure time, with the most prominent decrease associated with RTDs printed using the mono-EG ink. For RTDs printed using the mono-EG ink, the nominal resistance decreased exponentially from 1576 Ω after 4 min of plasma exposure to 75 Ω after 22 min, with the resistance essentially stabilizing after 16 min. Previously, our group characterized Ag mono-EG samples before and after plasma treatment by SEM. High-magnification SEM images revealed a dense, continuous morphology of the as-printed samples, whereas samples after plasma treatment showed a percolated network, signifying the sintering of converted Ag particles. The decrease in resistance as a result of plasma conversion is attributed to a change in chemical composition from insulating AgNO_3_ to conductive Ag, as well as a change in morphology [11]. The sensors made using the di-EG ink also exhibited a nonlinear decrease in resistance with increasing plasma exposure from an average of ~249 Ω at 15 min to ~57 Ω at 30 min, with the resistance essentially stabilizing after 20 min. For sensors printed using the tri-EG ink, the resistance decreased linearly with increasing plasma exposure from an average of ~244 Ω at 15 min to ~63 Ω at 30 min. These results are consistent with the sheet resistance measurements for strain gauges made using mono-EG, di-EG, and tri-EG inks reported previously [10]. For RTDs made using the di-EG and tri-EG inks, plasma exposure times less than 15 min resulted in structures with resistances that exceeded the measurement range of the multimeter (100 MΩ) and, thus, were excluded from further study. Figure 5b presents the resistance versus plasma exposure time for the same sensors measured at 59 °C. A comparison of the data profiles in Figure 5 and Figure 6 indicates that the printed structures were morphologically stable within this temperature range.

### 3.2. Temperature Coefficient of Resistance

To compare the performance of the RTDs printed using each of the three inks, the TCR was determined by applying the following equation to measurements of resistance versus temperature:(1)$$\mathrm{TCR}=\frac{∆R}{{\;R}_{o}}\frac{1}{\;(T–T_{o})\;}=\frac{∆R}{{{\;R}_{o\;}}_{\;}}\frac{1}{\;∆T}\;({}^{\circ}C{\;}^{-1})$$
where${\;R}_{o\;}$is the measured resistance at the reference temperature${\;T}_{o\;}$, ∆R is the difference in resistance between $T_{o\;}$and a temperature $T_{\;}$, and ∆T is the difference in temperature between${\;T}_{o\;}$and T. For materials governed by Equation (1), the slope of a plot of ∆*R*/$R_{o}$versus ∆*T* is the TCR.

Figure 6a presents ∆*R*/$R_{o}$ versus ∆T for sensors printed using the Ag mono-EG ink. For all plasma exposure times, ∆*R*/$R_{o}$exhibits a strong linear relationship with ∆T. Moreover, the slope increases from 5.32 × 10^–4^/°C to 1.58 × 10^−3^/°C as the plasma treatment time increases from 4 to 13 min, whereas for treatment times between 16 and 22 min, the slopes are essentially the same at ~1.50 × 10^−3^/°C. Figure 6b,c present ∆*R*/$R_{o}$ versus ∆T for sensors printed using the di-EG and tri-EG inks, respectively. Similar to the RTDs printed using the mono-EG ink, these plots also exhibit a strong linear dependence with temperature. However, the spread of TCR values is greatly smaller, ranging from 6.13 × 10^−4^/°C to 8.09 × 10^−4^/°C for sensors printed using the di-EG ink and 2.92 × 10^–4^/°C to 7.27 × 10^−4^/°C for sensors printed using the tri-EG ink.

Figure 7 summarizes the TCR versus plasma exposure time for RTDs fabricated using the mono-EG, di-EG, and tri-EG inks. Each data point represents the average TCR value from three test specimens, and the error bars represent the standard deviation of these measurements. As seen in Figure 7, RTDs printed using the mono-EG ink exhibit a steady increase in TCR from 5.32 × 10^−4^/°C after 4 min of plasma exposure to nearly 1.7 × 10^−3^/°C after 13 min, then maintains a relatively constant value of nearly 1.5 × 10^−3^/°C for plasma exposures between 15 and 22 min. In contrast, the TCR values for sensors printed using the di-EG and tri-EG inks were about a factor of 2 to 3 lower than sensors printed using the mono-EG ink at about 7.23 × 10^−4^/°C for sensors printed using the di-EG ink and about 4.88 × 10^−4^/°C for sensors fabricated using the tri-EG ink. For both di-EG and tri-EG inks, the TCR values were relatively constant over the entire range of plasma treatment times, as opposed to the mono-EG, which increased by one order of magnitude between 4 and 13 min. It should be noted that for the di-EG and tri-EG inks, a plasma duration of less than 15 min yielded structures whose resistances were too high to be measured, presumably because continuous Ag films were not fully formed under such conditions. The highest TCR evaluated in this study was at (~1.7 × 10^−3^/°C) for a sensor printed using the mono-EG ink after plasma exposure for 13 min and a substrate temperature of ~92 °C. This TCR is comparable to values reported for sensors made using Ag nanoparticle inks but was achieved at a much lower substrate temperature.

### 3.3. Sensitivity

To further compare the performance of the RTDs printed using each of the three inks, the sensitivity of each sensor was determined by applying the following equation to the room temperature resistance measurements presented in Figure 5a and the corresponding TCR measurements in Figure 7:

(2)$$\mathrm{Sensitivity}=TCR\times R_{o}=\frac{∆R}{R_{o\;}}\frac{1}{∆T\;}R_{o\;}=\frac{∆R}{∆T\;}\;(\Omega /{}^{\circ}C$$
where$\;R_{o}$ is the measured resistance at the reference temperature${\;T}_{o}$, ∆R is the difference in resistance between${\;T}_{o}$ and a temperature T, and ∆T is the difference in temperature between${\;T}_{o}$ and T. Figure 8 presents a plot of RTD sensitivity versus plasma exposure time for devices printed using the three inks. For devices printed using the di-EG and tri-EG inks, the sensitivity decreases monotonically from roughly 0.2 to 0.03 Ω/°C for plasma exposure times between 15 and 30 min. For devices printed using the mono-EG ink, the sensitivity also decreases with respect to plasma exposure time but over a much larger range, from 0.84 to 0.11 Ω/°C for plasma exposure times between 4 and 22 min, with the largest drop between 4 min and 7 min. The sensitivity measured for the sensor exposed to plasma for 4 min is the highest among the RTDs described in Table 2 but produced at a much lower temperature (~35 °C).

### 3.4. SEM Analysis

Figure 9 presents representative plan-view SEM images of RTDs printed using the three inks after plasma treatment. These images illustrate the strikingly different microstructures associated with the three inks. Figure 9a, which is an SEM image taken from an RTD printed using the mono-EG ink, shows a relatively smooth surface and dense microstructure as compared with the much rougher, semi-porous surface of the sensor printed using the di-EG ink (Figure 9b). In stark contrast, Figure 9c shows that the samples printed using the tri-EG ink exhibit a textured surface that is superposed onto an undulant, macro-porous 3D structure that extends into the bulk.

## 4. Analysis and Discussion

In previous work by our group, we found that the porosity and roughness of plasma-treated AgNO_3_/ethylene-glycol-based inks exhibited a noticeable dependence on the vapor pressure of the type of ethylene glycol solvent, namely mono-EG, di-EG, and tri-EG [13]. Specifically, the surface roughness and porosity increased with a decrease in the vapor pressure of the glycol solvent. Moreover, the resistivity increased as surface roughness and porosity increased. Structures printed using the Ag/tri-EG ink had the highest porosity and highest surface roughness and, as a result, produced the highest resistivity [13].

The information presented in Figure 7 and Figure 9 indicates a connection between the microstructure of the RTDs printed using the three ethylene-glycol-based inks and their TCR values. The highest TCR was observed in sensors printed using the mono-EG ink, which exhibited a smooth, nonporous texture. In contrast, the TCR values for the rough, porous sensors printed using the di-EG and tri-EG inks were considerably lower, with the highest TCR value associated with these sensors being 2.25 to 3.5 times lower than the TCR for the sensors printed using the mono-EG ink. Additionally, the degree of surface texture and porosity also seemed to influence the TCR, as the sensors printed using the di-EG ink had a moderately larger TCR than the sensors printed using the tri-EG ink (~15%). These observed differences in TCR cannot be attributed to gross differences in electrical resistance since the comparison set (plasma exposure times > 15 min) consisted of sensors that exhibited comparable room temperature resistances, thereby making microstructure the most significant difference between the sensors. A connection between TCR and porosity was previously reported for temperature sensors fabricated using spray-coated Ag nanowire inks [19]. In that study, TCR was observed to increase with increasing Ag nanowire density. Moreover, TCR was further increased by thermal annealing (up to 200 °C), which densified the nanowire network to form a thin-film-like structure comprised of a network of interconnected nanowires. The authors attributed TCR values lower than bulk Ag to voids in the nanowire network and polymer residuals from the ink. We believe that our findings are consistent with those reported in this study, namely that porosity exerts a strong negative influence on TCR.

Unlike the RTDs printed using the di-EG and tri-EG inks, RTDs fabricated using the mono-EG ink exhibited TCR values that increased from 5.32 × 10^−4^/°C to 1.7 × 10^−3^/°C as plasma exposure times increased from 4 to 13 min before stabilizing at ~1.5 × 10^−3^/°C for longer plasma treatment times. This behavior corresponds to a significant drop in nominal resistance from 1576 Ω to 141 Ω between 4 and 13 min before reaching a consistent value of ~76 Ω for longer plasma exposure times. This correlation between TCR and resistance for these RTDs may also relate to microstructure when considering the mechanism by which plasma exposure leads to the formation of Ag structures. As described previously and illustrated in Figure 2, the ethylene glycol solvent promotes the dissociation of AgNO_3_ into Ag^+^ and NO_3_^–^, which remain suspended in the solvent. Solvated electrons that originate in the Ar plasma reduce the Ag ions, leading to the formation of neutral Ag nuclei, which grow during plasma exposure. Concurrently, ethylene glycol evaporates during plasma bombardment in the low-pressure environment, as do NO_3_^–^ ions. Ag structures continue to grow until all of the ethylene glycol has evaporated or plasma exposure is terminated. Ag ions (Ag^+^) that do not react with solvated electrons instead react with NO_3_^–^ ions to form AgNO_3_ that concentrate away from the plasma near the substrate. The stability of the solvent during plasma exposure strongly influences this complex process. As shown in Table 1, mono-EG had the highest vapor pressure and lowest boiling point of the three solvents and was, therefore, least stable under plasma treatment conditions. The relatively short lifetime of mono-EG during plasma exposure favored the formation of relatively small Ag nuclei, leading to the formation of a thin, dense, nonporous structure in the near-surface region of the printed structure via 2D growth. In contrast, the more stable di-EG and tri-EG solvents favored the formation of much larger Ag nuclei via 3D growth, leading to the formation of porous structures, with the porosity being highest for tri-EG, which had the lowest vapor pressure and highest boiling point of the three forms of EG. For the RTDs printed using the mono-EG ink, the resistance measurements in Figure 5 indicate that maximum film thickness was achieved at a plasma exposure time of around 13 min, where the resistance saturated at a minimum value. Between 4 and 13 min, the thickness of structures increased with increasing exposure time, which, in turn, was associated with decreasing resistance and increasing TCR until additional plasma exposure did not appreciably affect the properties of the printed structure because the reactions detailed in Figure 2 had essentially ceased.

Unlike TCR, microstructure did not appear to influence the sensitivity of the RTDs, at least for the devices in the comparison set (plasma exposure time ≥ 15 min). In fact, the sensitivity decreased with increasing plasma exposure time regardless of ink solvent, with values grouped in a range between 0.2 and 0.03 Ω/°C. Figure 8 shows that the highest sensitivities were observed for RTDs printed with the mono-EG ink for plasma exposure times that lie outside of the comparison set (<15 min). This observation can be understood by examining the resistance associated with these structures. Because the formation of Ag in structures printed using the mono-EG ink favored 2D growth due to the high vapor pressure of mono-EG, very thin, continuous structures began to form at plasma exposure times as short as 4 min. These thin structures exhibit a high resistance due to their small cross-sectional area combined with elevated resistivity [11]. Additional plasma exposure causes the thickness of these structures to increase, thus increasing the cross-sectional area, reducing resistivity [11] and, therefore, reducing the overall resistance. Consequently, for the thinnest RTDs printed using the mono-EG ink, the high sensitivity was a result of the high resistance dominating over a modest TCR. For thicker structures, the reduction in resistance dominated over the increase in TCR, leading to the observed reduction in sensitivity. For long plasma exposure times, the sensitivity eventually saturated because both the TCR and resistance reached stable values because the Ag structures reached maximum thickness.

Table 2 summarizes the fabrication techniques and performances of inkjet-printed RTDs. The sensitivity of the RTDs was scaled by dividing the size of the RTDs. The sensitivity per unit area of the mono-EG Ag RTD was 7.0 × 10^−3^/°C/mm^2^, which is only about 50% lower than the highest value reported by Mattana et al. [5]. The RTD performance was further improved by formulating the ink to achieve a higher printing resolution, which led to a higher nominal resistance resulting from narrower traces with narrower spacings. From a materials perspective, the optimal RTD material featured both a high resistivity and TCR. The TCRs of the plasma-converted RTDs ranged from 7.3 × 10^−4^/°C to 1.7 × 10^−3^/°C, which are comparable to those of the other inkjet-printed RTDs. The sensitivity was ultimately improved by increasing resistivity using different types of solvent and tuning plasma treatment time.

## 5. Summary and Conclusions

This paper presented the results of a study designed to explore the relationship between microstructure and TCR for inkjet-printed RTDs using AgNO_3_/ethylene glycol inks that differ by type of ethylene glycol used as the ink solvent. Testing of RTDs revealed that the highest TCR values were associated with devices printed using mono-EG ink, followed by di-EG ink, and then by tri-EG ink. Testing also revealed that the sensitivity was essentially the same for RTDs printed using all three inks except for devices printed using mono-EG ink and exposed to plasma for short durations. SEM analysis showed that structures printed using the mono-EG ink exhibited a dense, nonporous microstructure after plasma treatment, whereas structures printed using di-EG and tri-EG ink had a porous microstructure with most porous structures associated with the tri-EG ink. Collective analysis of the TCR measurements and SEM imaging indicate a strong link between TCR and microstructure, with the TCR of the nonporous RTDs being more than three times larger than the most porous devices. These findings add to previous reports that found that microstructure strongly influences the performance of strain gauges and chemical sensors printed using these inks, with porosity enhancing the gauge factor of strain gauges and the sensitivity of hydrogen peroxide sensors. In contrast, analysis of the sensitivity measurements with respect to SEM imaging indicates that sensitivity is not influenced by porosity. The highest TCR and sensitivity for RTDs fabricated using the AgNO_3_/mono-EG ink are comparable to those reported using conventional Ag nanoparticle inks but produced at considerably lower substrate temperatures (between ~35 °C and 92 °C), making this process suitable for temperature-sensitive substrates.

## Figures and Tables

**Figure 1 micromachines-15-00749-f001:**
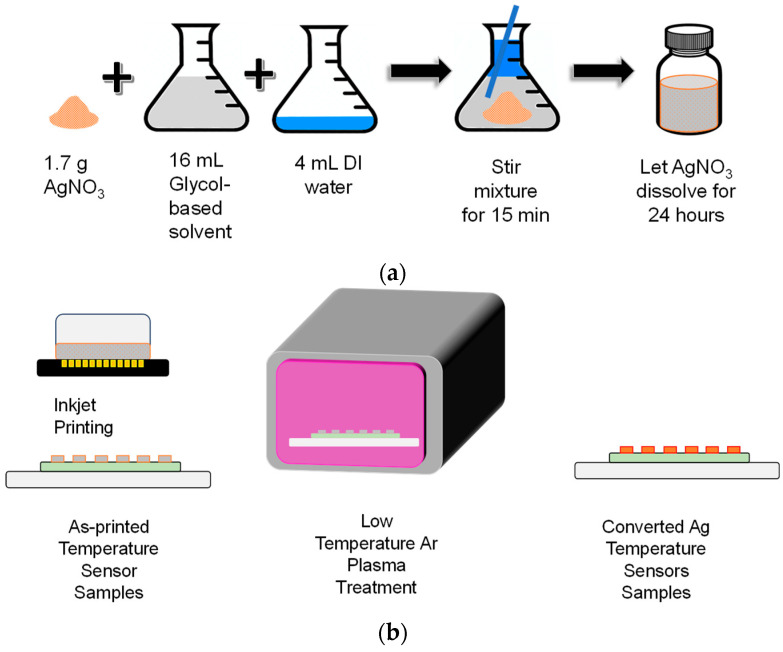
Schematic diagram of the temperature sensor fabrication steps: (**a**) ink preparation, (**b**) inkjet printing and plasma exposure.

**Figure 2 micromachines-15-00749-f002:**
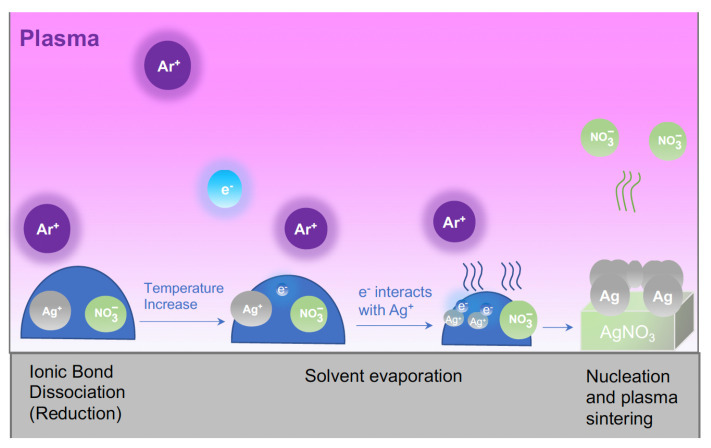
A schematic diagram showing the key steps associated with low-pressure treatment of AgNO_3_/ethylene glycol inks.

**Figure 3 micromachines-15-00749-f003:**
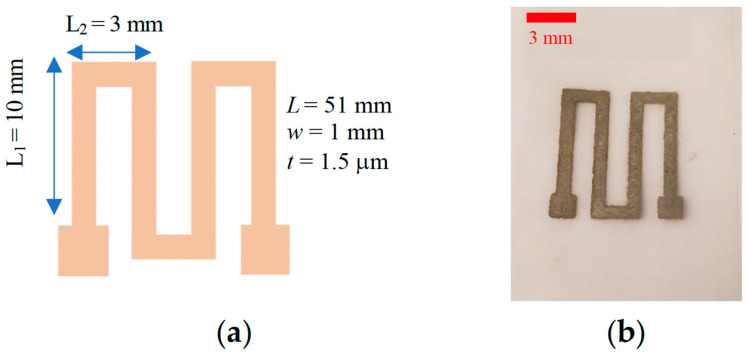
(**a**) A schematic diagram of the RTD. (**b**) Plan-view photograph of an inkjet-printed RTD after plasma treatment.

**Figure 4 micromachines-15-00749-f004:**
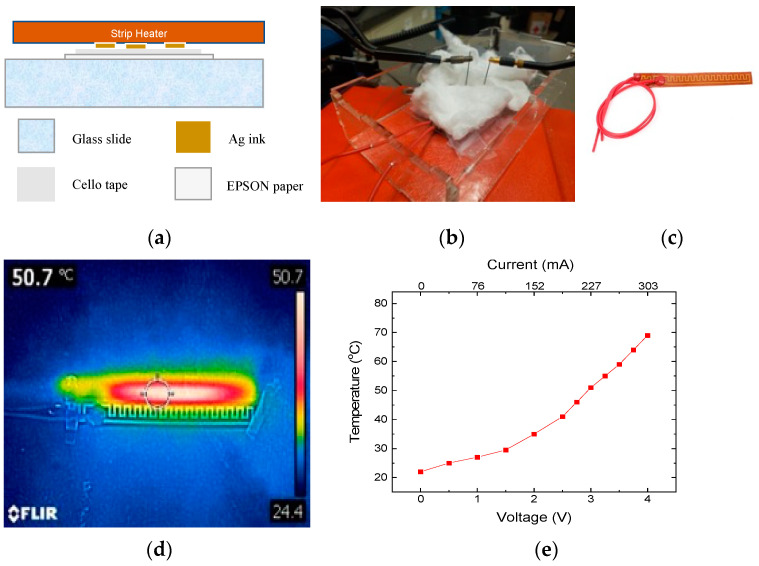
RTD testing: (**a**) Cross-sectional view of the testing setup. The temperature sensor was affixed to a glass slide. (**b**) Photograph of the custom-built rig, (**c**) Photograph of the flexible polyimide strip heater, (**d**) Thermal infrared image of the heater at ~51 °C, and (**e**) Temperature versus voltage profile for the strip heater.

**Figure 5 micromachines-15-00749-f005:**
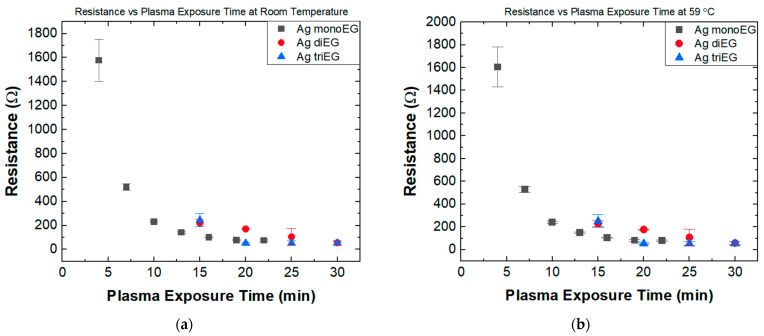
(**a**) Resistance versus plasma exposure time at room temperature for temperature sensors fabricated using the three EG inks; (**b**) Resistance versus plasma exposure time at 59 °C for temperature sensors fabricated using the three EG inks.

**Figure 6 micromachines-15-00749-f006:**
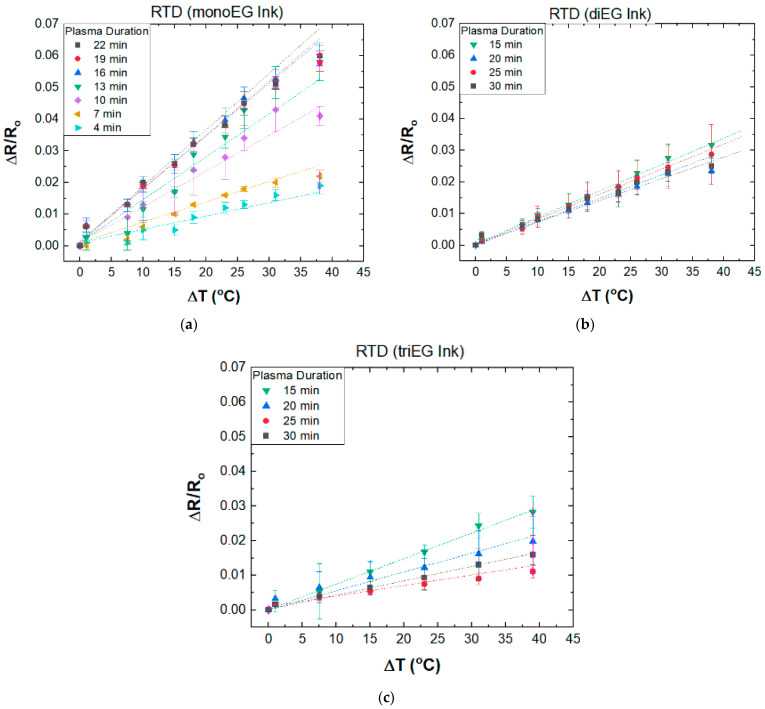
(**a**) Normalized change in resistance versus change in temperature for RTDs printed using the mono-EG ink. The plasma exposure time for each RTD is listed in the inset; (**b**) Normalized change in resistance versus change in temperature for RTDs printed using the di-EG ink. The plasma exposure time for each RTD is listed in the inset; (**c**) Normalized change in resistance versus change in temperature for RTDs printed using the tri-EG ink. The plasma exposure time for each RTD is listed in the inset.

**Figure 7 micromachines-15-00749-f007:**
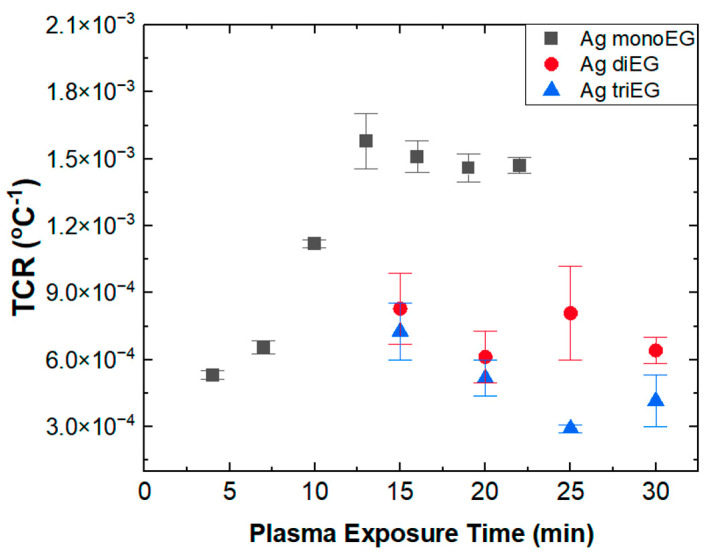
TCR versus plasma exposure time for RTDs fabricated using the mono-EG, di-EG, and tri-EG inks.

**Figure 8 micromachines-15-00749-f008:**
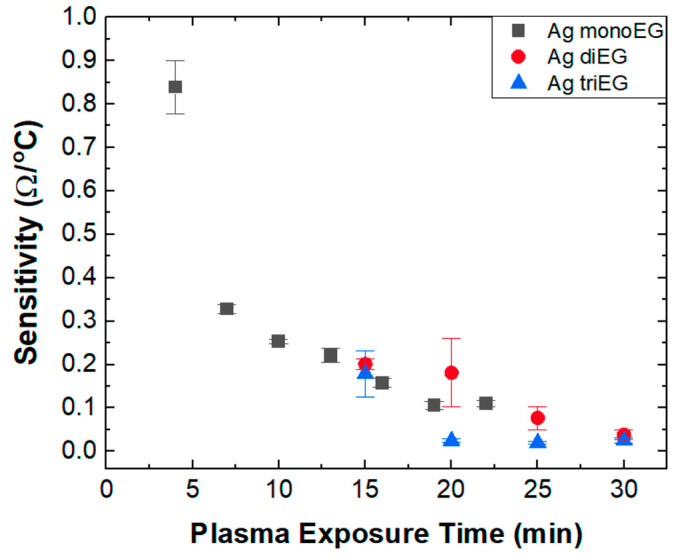
Sensitivity versus plasma exposure time for RTDs fabricated using the mono-EG, di-EG, and tri-EG inks.

**Figure 9 micromachines-15-00749-f009:**
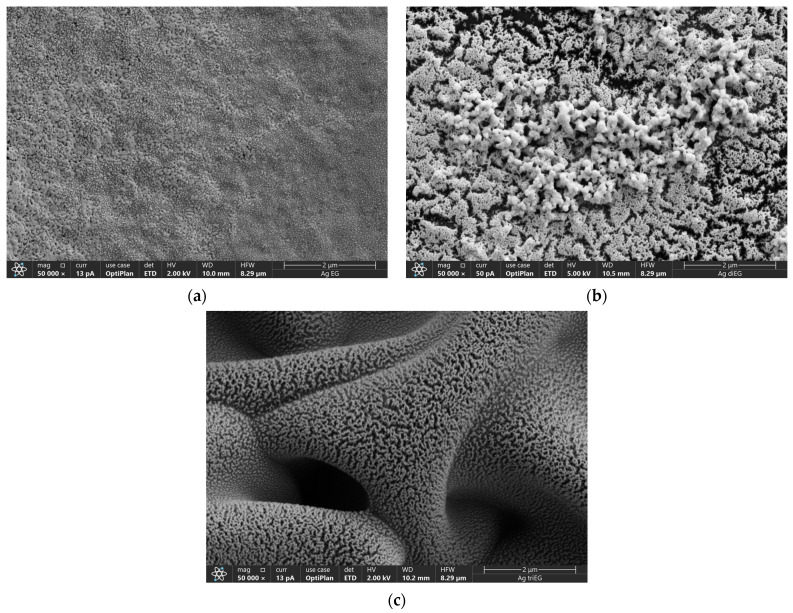
Representative plan-view SEM images from sensors fabricated using the three inks: (**a**) mono-EG ink after 30 min of plasma exposure, (**b**) di-EG ink after 30 min of plasma exposure, and (**c**) tri-EG ink after 30 min of plasma exposure.

**Table 1 micromachines-15-00749-t001:** Properties of mono-EG, di-EG, and tri-EG.

Organic Solvent	Vapor Pressure (mm Hg)	Viscosity (cP)	Boiling Point (°C )	Density (g/cm^3^)	Ref.
Monoethylene Glycol	0.12 at 20 °C	21	196	1.112	[14]
Diethylene Glycol	0.006 at 20 °C	38	245	1.118	[15,16]
Triethylene Glycol	0.00075 at 25 °C	48	285	1.120	[17]

**Table 2 micromachines-15-00749-t002:** A summary of reported Ag-based RTDs printed using Ag nanoparticle inks compared with this work.

Ref.	Particle Size	Sintering Temperature	Substrate	Operating Temperature (°C)	Nominal Resistance (Ω)	TCR (°C ^−1^)	Area (mm^2^)	Sensitivity/Area (Ω/°C/mm^2^)
(Barmpa., 2018) [1]	20 nm	Let dry after printing for 14 days	Glossy photo paper	25–75	N/A	9.06 × 10^−4^	N/A	N/A
(Courbat, 2011) [2]	≤50 nm	@150 °C for 60 min	Raw paper coated with oxide film	−20–60	600–710	1.1 × 10^−3^	256	3.1 × 10^−3^
(Felba, 2009) [3]	6 nm	Sintered at 250 °C for 1 h	Glass slides	20–200	1	2.08 × 10^−3^	28.2	7.4 × 10^−5^
(Dankoco, 2016) [4]	N/A	**First step**: drying solvent 130 °C for 10 min **Second step**: 150 °C for 30 min	Kapton^®^ (polyimide)	20–60	2 k	2.27 × 10^−3^	6441	7.1 × 10^−4^
(Mattana, 2013) [5]	<150 nm	Annealed at 290 °C for 30 min	Kapton^®^	10–80	52	2.99 × 10^−3^	10.2	1.5 × 10^−2^
(Ali,2016) [6]	115 nm	130 °C for 30 min	PET	0–100	72–341	1.08 × 10^−3^	49	7.5 × 10^−3^
(Khalaf, 2022) [7]	N/A	140 °C for 10 min	Kapton^®^ (polyimide)	28–90	7–23	1.77 × 10^−3^	100	4.1 × 10^−4^
(Zikulnig, 2019) [8]	N/A	Photonic curing	Paper	20–80	0.5 k–7 k	1.63 × 10^−3^	1147	1.0 × 10^−2^
This Work	N/A	Plasma conversion	Cellophane	20–60	75–1567 _(mono-EG)_ 57–220 _(di-EG)_ 53–244 _(tri-EG)_	1.7 × 10^−3^ _(mono-EG)_ 8.1 × 10^−4^ ^(di-EG)^ 7.3 × 10^−4^ ^(tri-EG)^	120	7.0 × 10^−3^ _(mono-EG)_ 1.7 × 10^−3^ ^(di-EG)^ 1.5 × 10^−3^ ^(tri-EG)^

## Data Availability

All data to support the results of this study are included in this article.

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
