# Peer review of "The Influence of Microstructure on TCR for Inkjet-Printed Resistive Temperature Detectors Fabricated Using AgNO_3_/Ethylene-Glycol-Based Inks"

_micromachines, 2024, doi:10.3390/mi15060749_

Round 1

Reviewer 1 Report

Comments and Suggestions for Authors

In the manuscript of micromachines-2988837, the authors investigated Ag inkjet-printed, resistive temperature sensors fabricated using particle-free inks based on a silver nitrate (AgNO3) precursor and three ethylene glycols (mono-EG, di-EG and tri-EG) as the ink solvent, and subjected to a low pressure Ar plasma after printing. The research did not offer new insight, comparing the previous published paper (Y. Sui, L. P. Kreider, K. M. Bogie, and C. A. Zorman, "Fabrication of a silver-based thermistor on flexible, temperature sensitive substrates using a low-temperature inkjet printing technique," IEEE sensors letters, vol. 3, no. 2, pp. 1-4. 2475- 472 1472, 2019.), so I recommend the mamuscript to be rejected, which has serious flaws. The following comments should be seriously concerned, and additional experiments are needed.

1. The as-printed samples were exposed to a low pressure Ar plasma using a 300 W, 13.56 MHz, March PX250 plasma system. Sample temperature rising problem (may lead to the decomposition silver nitrate) during the RF sputtering should be discussed in detail, and the etching of AgNO3, Ag NP, and  three ethylene glycols should be concerned.

2. Could the ultraviolet light along with the production of plasma be the cause of the decomposition of AgNO3?

3. As the plasma is a surface treating technique, XPS measurement should be added to analyse the elements and their valence states.

4. Silver's conversion ratio of ink along the thinkness should be analysed and discussed.

5. As for temperature sensing, the following key parameters and characteristic should carefully measured, including the response time, recovery time, low-high temperature cycle testing, low-high humidity cycle testing.

6. Based on the results, the research motivationthree of three ethylene glycols (mono-EG, di-EG and tri-EG) is confusing, which should be clarified.

7. The influence of substrate‘s thermal expansion coefficient on device's temperature sensitivity should be considered.

Author Response

Please find attached the pdf file.

Reviewer 2 Report

Comments and Suggestions for Authors

The research work “Inkjet Printed Resistive Temperature Detectors Fabricated using 2 AgNO3/Ethylene Glycol-Based Inks: The Influence of Micro- 3 structure on TCR” submitted by Aziz et al. demonstrates an extensive study of RTD based on silver nitrate precursor dissolved in ethylene glycol. The study shows good scientific analysis and experimental attributes. I hereby recommend acceptance of the manuscript. However, I would like the authors to address the below two questions:

The author mentions “The printer drop spacing was set at 15 µm and the platen temperature was set at 50 oC for Ag/mono-EG, 43 oC for Ag/di-EG and 41 oC for Ag/tri-EG inks to ensure the production of uniform patterns on the cellophane substrates”.

The temperature of the platen may also have some role to play in the porosity and RTD performance. Why is the temperature of the platen different for mono, bi and tri ethylene glycol?

Did the authors work upon understanding the porosity effect due to platen temperature?

Author Response

Please find attached the pdf file.

Reviewer 3 Report

Comments and Suggestions for Authors

Dear Editor,

I have carefully reviewed this paper and find it to have an interesting idea suitable for publication in the Journal of Micromachines. However, there are several issues that need to be addressed before acceptance:

-      It's suggested to include Schematic 12 (formation of Ag structures) within Fig. 1 (Schematic diagram of the temperature sensor fabrication steps) and discuss it here. This will enhance understanding of the experimental process and formation mechanism of microstructure due to plasma.

-      SEM images prior to the plasma are crucial to understanding how plasma affects texture and roughness. I recommend adding SEM images for the as-printed sample as well and then updating the related discussion accordingly.

-      In the Resistance versus Plasma Exposure Time section, it's crucial to offer concise scientific reasons for why Ag/monoEG shows lower resistance compared to the other two RTDs before and during different plasma treatment intervals. If authors suggest attributing this to the smooth surface of the Ag/monoEG-based sample, it's vital to include SEM images (as recommended in comment 2) before and after plasma treatment.

-      In Fig. 4 (Resistance versus Plasma Exposure Time at room temperature) and Fig. 5 (Resistance versus Plasma Exposure Time at 59°C), it may be understandable that each device exhibits similar behavior at different heating conditions (room and 59°C). However, it's important to address why each sample shows similar resistance, such as Ag/monoEG:  1600 Ω, at both 25°C and 59°C.

-      It's noted that the authors didn't obtain resistance responses from the samples (di-EG and tri-EG) due to limitations of the multimeter (100 M Ω) (line: 162-164 and 262-264). However, it's recommended to test and validate their results using another device, such as a Keithley meter with a micro-ohm/nano-ohm range. This is crucial for journal publication as it ensures the clarity of their work and validates the scientific conclusions drawn from the data.

-      Consider arranging figures more compactly, such as combining Fig. 4 and Fig. 5 as (a) and (b), and Fig. 6 to Fig. 9 as (a-d), to facilitate understanding and accessibility of the target theme from figures.

Comments on the Quality of English Language

Minor editing of English language required

Author Response

Please find attached the pdf file.

Round 2

Reviewer 1 Report

Comments and Suggestions for Authors

Dear authors,

You have well addressed my comments, and the manuscript had been significantly improved; so I recommend the revised manuscript to be accepted in Micromachines.

Author Response

We thank the reviewer for re-reviewing our manuscript and recommending it for publication.

Reviewer 3 Report

Comments and Suggestions for Authors

The authors have addressed my comments in their response letter; however, the corresponding discussion was not included in the manuscript.

As the reviewer's questions indicate that similar questions may arise in the reader's mind. Therefore,  before accepting this paper, I request the authors to do the following tasks:

1. Incorporate a detailed discussion of your responses to my comment # 2 and #3 in the sections where these comments were raised.
If you are referring to any reference papers, kindly discuss them with brief details in the corresponding sections. 

2. Please proofread and refine the English with the help of a native speaker or an editing company to ensure that the reader can promptly understand your ideas and discussions.

Thank you.

Author Response

The authors have addressed my comments in their response letter; however, the corresponding discussion was not included in the manuscript.

As the reviewer's questions indicate that similar questions may arise in the reader's mind. Therefore, before accepting this paper, I request the authors to do the following tasks:

1. Incorporate a detailed discussion of your responses to my comment # 2 and #3 in the sections where these comments were raised. If you are referring to any reference papers, kindly discuss them with brief details in the corresponding sections.

We thank the reviewer for this reminder. Response to comment#2 has been added in the manuscript (starting from page 5 line 172). Response to comment #3 has been included on page 12, line 374.

2. Please proofread and refine the English with the help of a native speaker or an editing company to ensure that the reader can promptly understand your ideas and discussions.Thank you.

We thank the reviewer for pointing this out. A proofreading tool was used to check the typos and grammar errors. Corrections were made accordingly.
